



# Fire in lichen-rich subarctic tundra changes carbon and nitrogen cycling between ecosystem compartments but has minor effects on stocks

Ramona J. Heim[1], Andrey Yurtaev[2], Anna Bucharova[1,3], Wieland Heim[1,4], Valeriya Kutskir[2], Klaus-Holger Knorr[5], Christian Lampei[1], Alexandr Pechkin[6], Dora Schilling[1], Farid Sulkarnaev[2], and Norbert Hölzel[1]

[1]Institute of Landscape Ecology, Biodiversity and Ecosystem Research Group, University of Muenster, Heisenbergstraße 2, 48149 Münster, Germany
[2]Research Institute of Ecology and Natural Resources Management, Tyumen State University, 6 Volodarskogo Street, Tyumen, Russia
[3]Department of Biology, Conservation Biology Group, University of Marburg, Karl-von-Frisch-Straße 8, 35043 Marburg, Germany
[4]Department of Biology, Animal Ecology, University of Turku, Vesilinnantie 520500, Turku, Finland
[5]Institute of Landscape Ecology, Ecohydrology and Biogeochemistry Group, University of Muenster, Heisenbergstraße 2, 48149 Münster, Germany
[6]Research Center of the Yamal-Nenets Autonomous District, Salekhard 629008, Russia

**Correspondence:** Ramona Heim (ramona.heim@uni-muenster.de)

**Abstract.** Fires are predicted to increase in Arctic regions due to ongoing climate change. Tundra fires can alter carbon and nutrient cycling and release a substantial amount of greenhouse gases with global consequences. Yet, the long-term effects of tundra fires on carbon (C) and nitrogen (N) stocks and cycling are still unclear. Here we used a space-for-time approach to investigate the long-term fire effects on C and N stocks and cycling in soil and aboveground living biomass. We collected data

from three large fire scars (>44, 28 and 12 years old) and corresponding control areas and used linear mixed-effects models in a Bayesian framework to analyse how the stocks and cycling were influenced by fire. We found that tundra fires did not affect total C and N stocks because a major part of the stocks was located belowground in soils, which were largely unaltered by fire. However, fire had a strong effect on stocks in the aboveground vegetation, mainly due to the reduction of the lichen layer. Fire reduced N concentrations in graminoids and herbs on the younger fire scars, which affected respective C/N ratios and indicated

an increased post-fire competition between vascular plants. Aboveground plant biomass was depleted in $^{13}$C in all three fire scars. This could be related to a lower $^{13}$C abundance in $CO_2$ in the ambient air because of increased post-fire decomposition, providing a source of $^{13}$C-depleted $CO_2$. In soil, the relative abundance of $^{13}$C changed with time after fire because of the combined effects of microbial decomposition and plant-related fractionation processes. Our results indicate that in lichen-rich subarctic tundra ecosystems, the contribution of fires to the release of additional carbon to the atmosphere might be relatively

small as soil stocks appear to be resilient.



## 1 Introduction

Arctic regions are warming faster than the worldwide average, with strong impacts on the global carbon cycle (Post et al.,
2019; Schuur et al., 2015). Warming is expected to expose ecosystems at high latitudes to increased fire frequency and extent
(Chen et al., 2021; Hu et al., 2015; Moritz et al., 2012; Young et al., 2016). Wildfires in the Arctic can feed back on climate
change because they release carbon stored in the tundra ecosystems (Lasslop et al., 2020; Veraverbeke et al., 2021). This effect
may be substantial because Arctic and sub-arctic regions store huge amounts of the global carbon (C) (ca. 1,700 billion tons
in terrestrial soils, (Schuur et al., 2015). Wildfires enhance the release of stored carbon from active ecosystems in multiple
ways. Combustion of living and dead biomass rapidly transfers large amounts of stored C to the atmosphere (Mack et al.,
2011). Wildfires also burn the insulating and reflecting layers of soil organic material and vegetation, which increases soil
temperatures through irradiation (Chambers et al., 2005; Jiang et al., 2015). Warmer soil temperatures raise microbial activity,
which in turn promotes faster decomposition processes (Jansson and Hofmockel, 2020). Soil warming induces permafrost thaw
and active layer deepening, which exposes organic matter in deeper layers to decomposition, resulting in a massive release of
carbon to the atmosphere (Estop-Aragonés et al., 2020; Schuur et al., 2009). In summary, the rate of C loss through fires is
substantial and faster than other climate-driven processes influencing C cycling in northern latitudes (Mack et al., 2011). Fires
also affect the stored N, which has consequences for vegetation and ecosystem functioning, because N limits productivity in
tundra ecosystems (Oulehle et al., 2016). Fires release substantial parts of the accumulated nitrogen (N) pool to the atmosphere.
However, soil warming and related deeper thaw may increase plant-available inorganic N through higher mineralisation rates
(Aerts, 2006; Salmon et al., 2016). This increases the productivity of shrubs and graminoids because they can reach the
nutrients mineralised in deeper soil layers than cryptogams (Dormann and Woodin, 2002; Oulehle et al., 2016; Salmon et al.,
2016). As a consequence, the cover of vascular plants increases after fires at the expense of previously dominating cryptogams,
which cannot profit from the enhanced nitrogen availability as they do not reach deep soil layers or cannot take up nitrogen
from the soil (Bret-Harte et al., 2013; Jandt et al., 2008; Narita et al., 2015; Turetsky et al., 2012). Such profound changes in
vegetation cover, in turn, influences ecosystem functioning and processes, such as carbon and nitrogen cycling and C and N
stocks (Longton, 1997; Sancho et al., 2016; Turetsky, 2003). Studies including nutrient cycles in soil and vegetation related to
long-term effects of tundra fires are rare and, therefore, the effect of fires on the carbon cycle of tundra ecosystems is relatively
unknown (Mack et al., 2011). This is unfortunate because the reactions of the tundra ecosystem to altered temperature have
a time lag and are thus relatively slow (Rinnan et al., 2007). The effects of fire on the functioning of the tundra ecosystem
may thus be visible only after more than a decade (Blok et al., 2018). Therefore, long-term studies are essential (Chapin Iii
et al., 2000). One widely used tool to unravel C and nutrient cycling processes is the use of stable isotopes (Peterson and Fry,
1987). The relative abundance of the heavy carbon isotope ($^{13}$C) in the living aboveground biomass and in the soil can provide
insight into post-fire environmental site conditions. In soil, increased activity of microbes after fire is reflected in the residual
enrichment of $^{13}$C in the remaining soil organic matter, as decomposers prefer isotopically light material ($^{12}$C) (Ehleringer





et al., 2000). In the aboveground biomass, the isotope ratio is related to the isotopic composition of the source $CO_2$, and thus
the isotope ratio in leaves may reflect the lower levels of $^{13}C$ in $CO_2$ which is available for photosynthesis through respiration
from soils during decomposition (Dawson et al., 2002; Farquhar et al., 1989). The isotope ratios of N can be used for a better
understanding of post-fire N cycling processes by providing information about the association with mycorrhizal fungi. During
N transfer from mycorrhizal fungi to the plant, the heavy $^{15}N$ gets enriched in the mycorrhizas and depleted in the plants.
During succession, the associated mycorrhizal fungi have to colonize the plants first, and therefore $^{15}N$ would be expected to
be enriched in early successional stages and depleted in later successional stages (Hobbie et al., 2005). Here, we studied the
long-term effect of fire on tundra nutrient stocks and cycling using a space-for-time approach. We focused on three large fire
scars (>44, 28, and 12 years old) that represent a gradient of post-fire succession. We analysed C and N stocks, concentrations,
and isotope ratios in vegetation and soil. We hypothesized that: 1) Fire leads to decreased ecosystem C and N stocks 2) Fire
affects long-term C and N cycling, evident through a) an increased N concentration and a decreased C/N ratio in vascular plants
after fire b) a depletion of $^{13}C$ in aboveground biomass and a relative enrichment of $^{13}C$ in soil after fire c) increased relative
abundance of $^{15}N$ in leaves of vascular plants on the younger fire scars

## 2   Material and Methods

### 2.1   Study area

The study area was located in Western Siberia within the Yamalo-Nenets Autonomous Okrug (Region) between the rivers Pur
and Taz, north of the Arctic Circle (centre at 67° 1'19.59"N, 79° 1'53.53"E, total study area size ca. 70 km$^2$) (Heim et al.,
2021). The region has a subarctic climate with a growing season from mid-June to early September, and with a mean annual
temperature of -8.1 °C, mean January temperature of -26.2 °C, mean July temperature of 14.4 °C, and annual precipitation of
482 mm (Kazakov, 2019). The vegetation in the region represents the transition zone between the forest-tundra to the south
and the shrub-tundra to the north (Yurkovskaya, 2011). Compared to other previously investigated tundra areas, e.g. moist
70   acidic tundra in the Anaktuvuk River region in Alaska (Mack et al., 2011), the region is a relatively dry and well-drained
upland area. The vegetation is dominated by reindeer lichens (mostly Cladonia spp., around 70% cover), with abundant shrubs
and dwarf shrubs, such as Betula nana (around 25% cover) and Vaccinium uliginosum (around 10% cover), and occasional
graminoids and bryophytes with low cover. The landscape is sparsely dotted with larch trees (Larix sibirica). Soils are Cryosols
(IUSS Working Group WRB, 2015), which developed in silty, loess-like parent material and soil organic layer thickness was
75   generally low at all sites. Bulk density ranged from 0.6 g/cm3 to 1.6 g/cm3 and pH from 3.8 to 6.3 (Fig. S1). Soil temperature
in 12 cm depth and permafrost thaw depth were higher on burnt sites but regenerated to control levels after > 44 years (Heim
et al., 2021). The main cause of fires in the study region is lightning (Kornienko, 2018), but the number of human-induced fires
increases because of expanding transport and settlement infrastructure due to oil and gas exploitation (Mollicone et al., 2006;
Vilchek and Bykova, 1992; Yu et al., 2015).



## 2.2 Sampling design and data collection

To examine the long-term impact of fire on C and N cycling in soil and aboveground biomass, we studied three large fire scars (aged 12,28, and >44 years) and adjacent unburnt control sites. All three fire scars were located close to each other (< 10 km distance between scars) and therefore had similar environmental conditions. We detected the three fire scars visually with the help of satellite images. We used annual Landsat images back to the year 1985 in Google Earth Timelapse (Gorelick et al., 2017) and older Landsat images back to 1973, which we downloaded via the USGS earth explorer (U.S. Geological Survey, 2018) (for details see Heim et al., 2021). The oldest fire scar (3,500 ha) was already visible on the first satellite image from 1973 (Landsat1). The medium-aged scar (ca. 12,500 ha) burnt in 1990, and the youngest scar (542 ha) burnt in 2005. We collected samples in July 2017 at fire scars burned before 1973 and in 2005 (that is >44 and 12 years after fire, respectively) and in July 2018 at the area burnt in 1990 (28 years after fire). Environmental conditions of the tundra ecosystem are stable with low inter-annual variability (Dahl, 1975) and climatic conditions in the two sampling years were similar (Kazakov, 2019). It is thus unlikely that collecting in two subsequent years biased the results. We placed 10 sampling locations along the fire border of each fire scar (at least 300 m apart from each other). At each location, we placed one sampling plot in the burnt and one sampling plot in the unburnt site, resulting in a total of 60 plots (30 pairs). Each site included a 10 m by 10 m plot and a soil profile, located 2-5 m next to the plot. The plots at each location were placed as close to each other as possible, but at least 100 m apart (50 m minimum plot distance from the fire border, to avoid edge effects). We sampled aboveground biomass in five subplots of the 10 by 10 m plot. The subplots were situated in the four corners and in the centre of the plot. While the subplots for shrubs, herbs, and graminoids were squares of 0.3 m x 0.3 m, the subplots for lichens and bryophytes were 0.1 m x 0.1 m and were randomly placed within the 0.3 m x 0.3 m squares. Biomass of each type (shrubs, graminoids, herbs, bryophytes, lichens) was separately sampled and all subplots were pooled in one bag per type. Soil was sampled in layers ranging from 0-5, 5-30, and 30-60 cm depth. In each layer, we placed three cylinders (200 cm3) horizontally and distributed them over the whole range and sampled the soil of one layer in one plastic bag.

## 2.3 Laboratory analyses and stock calculations

In the field, aboveground biomass was stored well ventilated, while soil was stored in airtight plastic bags. We dried aboveground biomass and soil in the laboratory at 60 °C to a constant weight. For determination of C and N concentrations and stable isotopic composition, samples were ground in a ball mill (tungsten carbide cups, MM400, Retsch, Haan, Germany) and weighed into tin capsules for analyses on an elemental analyser (EA 3000, Eurovector, Padua, Italy) coupled to an isotope ratio mass spectrometer (EA-IRMS, NuHorizon, Nu Instruments, Wrexham, UK). Calibrations were done using certified working standards (IVA Analysentechnik, Meerbusch, Germany) and reference materials (IAEA 600 (Coplen et al., 2006), USGS 35 (Böhlke et al., 2003)). We could not analyse nitrogen isotopic composition in the soil samples, as the N concentrations were generally too low. We express the stable isotope composition using the common delta notation as a ratio relative to an internationally accepted reference standard: $\delta XXE = 1000 \cdot (R_{sample}/R_{standard} - 1)$, ‰. Where E is the element, xx the mass of the heavier isotope, and R is the abundance ratio of the isotopes (e.g., $^{15}N{:}^{14}N$) (Dawson et al., 2002). Higher $\delta$ values indicate a





higher abundance of the heavier isotope (Dawson et al., 2002). We furthermore analysed bulk density. We dried a subsample of soil samples at 105°C and calculated bulk density using the following formula: bulk density (g/cm3) = dry soil weight (g) / soil

volume (cm3). Based on the bulk density, we calculated soil stocks for each range with following formula: stocksoil (kg/m2) = bulk density (g/cm3) * concentration/100 * range size (cm) (e.g. 25 cm for 5-30 cm depth range) / 1000 (g/cm3 in kg/m2). C and N stocks for vegetation were calculated as follows: stockvegetation (kg/m2) = dry mass (kg) * sampling area in m2 * concentration/100

## 2.4  Statistical analysis

We carried out all statistics in R, Version 4.0.3. (R Core Team, 2020) and fitted all models in a Bayesian framework using the function brm() from the package brms (Bürkner, 2017, 2018). For testing anticipation 1 (Fire leads to decreased ecosystem C and N stocks), we summarised aboveground biomass and soil stocks per sampling site. We fitted two linear models with log-transformed stocks (C stocks, N stocks) as dependent variables. Fire scar (12, 28, >44 years), burn status (burnt, unburnt), and their interaction were included as independent variables and location (paired plots) as a random factor. To obtain the posterior

distribution, we used improper priors and ran 6000 iterations (warmup = 3000) with four chains. The posterior distribution is a probability distribution that summarises updated beliefs about the parameter after observing the data and is thus a result of the prior distribution and the likelihood function (Korner-Nievergelt et al., 2015). We present mean values together with the 95% credible interval (CrI) of the simulated posterior distribution. The 95% CrI is the range in which the true value is expected with a probability of 0.95. We calculated the posterior probabilities by using the proportion of simulated values of the posterior

distribution of C and N stocks on burnt sites being smaller than the proportion of simulated values of the posterior distribution of stocks on unburnt control sites. Therefore, a posterior probability of 1 would indicate that the stocks on the burnt sites were significantly lower than on unburnt control sites. A posterior probability of 0 would mean the opposite and a posterior probability of 0.5 indicates that stocks at burnt and unburnt sites did not differ. To analyse the impacts of fire on aboveground biomass and soil stocks separately, we fitted four linear models with log-transformed stocks (aboveground biomass C stocks,

aboveground biomass N stocks, soil C stocks, soil N stocks) as dependent variable. Fire scar (12, 28, >44 years), burn status (burnt, unburnt), and vegetation type for biomass (shrub leaves, herbs, graminoids, bryophytes, lichens) or depth for soil (0-5 cm, 5-30 cm, 30-60 cm) were included as independent variables with all possible interactions. Sites were nested in location as random factor. To obtain the posterior distribution, we used improper priors and ran 4000 iterations (warmup = 2000) with four chains. For the models of C/N ratio in biomass and C concentration in soil, we used adapt_delta=0.95 and

run 6000 iterations (warmup=3000) for better convergence. Bayesian $R^2$ was calculated with the function bayes_$R^2$ from the package brms (Bürkner, 2017, 2018). For testing the anticipation 2 (Fire affected long-term C and N cycling), we first calculated the differences between burnt and unburnt control plots (burnt - unburnt) for concentrations, ratios, and isotope ratios as this enabled us a more straightforward and clearer interpretation (see raw data in Fig, S2, Fig. S3). A negative difference of the C concentration, therefore, indicates that fire decreased the C concentration. A negative difference in the C/N

ratio indicates that fire decreased the C/N ratio – either through decreased C concentrations or increased N concentrations. A negative difference of $\delta^{13}$C ($\Delta^{13}$C) indicates that fire decreased $\delta^{13}$C (and thus caused a depletion of $^{13}$C). We fitted 9 linear





models with concentrations, C/N-ratio, and isotope ratios (C in biomass, C in soil, N in biomass, N in soil, C/N in biomass, C/N in soil, $\delta^{13}$C in biomass, $\delta^{13}$C in soil, $\delta^{15}$N in biomass) as dependent variable. Fire scar age (12, 28, >44 years), burn status (burnt, unburnt), and vegetation type for biomass (shrub leaves, herbs, graminoids, bryophytes, lichens) or depth for soil (0-5

cm, 5-30 cm, 30-60 cm) were included as independent variables with all possible interactions. Sites were nested in location as random factor. We controlled for non-constant variances in type and depth adapting sigma (e.g. sigma ~ 0 + type). To obtain the posterior distribution, we used improper priors and ran 4000 iterations (warmup = 2000) with four chains.

## 3    Results

### 3.1    Fire influence on C and N stocks in aboveground biomass and soil

Fire had no effects on total stocks (C: $P_{burnt\_12<control\_12} = 0.68$, $P_{burnt\_28<control\_28} = 0.94$, $P_{burnt\_44<control\_44} = 0.16$, N: $P_{burnt\_12<control\_12}$ = 0.40, $P_{burnt\_28<control\_28} = 0.81$, $P_{burnt\_44<control\_44} = 0.18$) (Fig. 1, Table S6, Table S7). The net losses/gains for the youngest fire scar were -2.17 kgC/m2 and 0.09 kgN/m2, for the intermediate -9.85 kgC/m2 and -0.33 kgN/m2 and for the oldest 5.41 kgC/m2 and 0.30 kgN/m2. Soils at burnt plots harboured less C at the intermediate fire scar in 5-30 cm depth. In the oldest fire scar, soils stored more N and C in the upper soil layer in comparison to the unburnt control (Table 1, Table S7, Table S8).

In contrast to soil, C and N stocks in vegetation were strongly reduced through fire (Fig. 1). This was mainly because fire substantially reduced lichens and consequently the C and N stored in them. The depletion of C and N in lichens lasted for more than four decades (Table 1, Table S7, Table S8). In contrast, fire increased C and N stocks in herbs in the intermediate fire scar and in bryophytes in the oldest one.





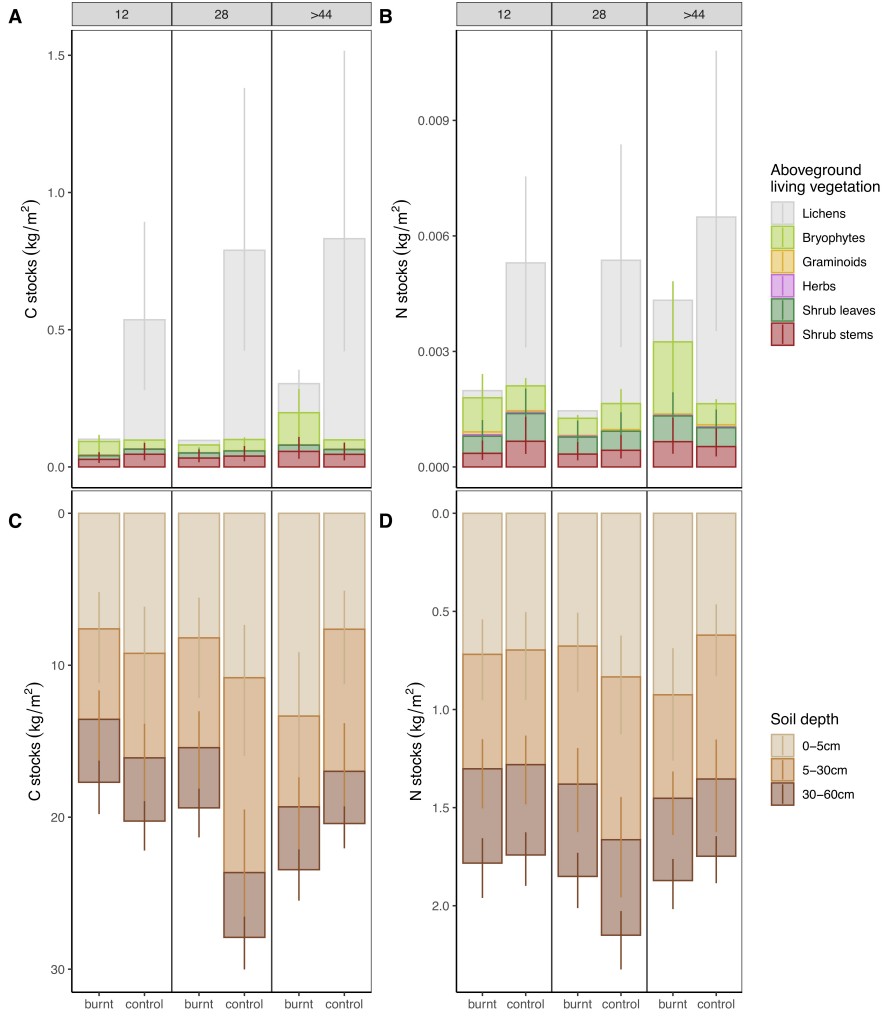

**Figure 1.** Predicted mean values for C and N stocks in aboveground biomass and soil in regard to time after fire on burnt and control plots. Vertical lines are 95% credible intervals (CrI). Model structure and parameters as well as corresponding CrI's and $R^2$ are shown in Table S6, Table S7 and Table S8 . C and N stocks for shrub stems are calculated as described in Figure S4. Compare A and B with aboveground dry weight in Figure S5.





**Table 1.** Probabilities of stocks on burnt plots to be lower than stocks on unburnt control plots. If probabilities are higher than 0.975 (stocks on burnt plots are lower compared to unburnt plots) or smaller 0.025 (stocks on burnt plots are higher) they are written in bold. If probabilities are 0.5 there is no difference between stocks on burnt plots and stocks on control plots.

| Vegetation Type | $C_{burnt} < C_{control}$ | | | $N_{burnt} < N_{control}$ | | |
|---|---|---|---|---|---|---|
| | 12 | 28 | >44 | 12 | 28 | >44 |
| Lichens | **1.00** | **1.00** | **1.00** | **1.00** | **1.00** | **1.00** |
| Bryophytes | 0.22 | 0.76 | **0.01** | 0.29 | 0.79 | **0.01** |
| Graminoids | 0.08 | 0.90 | 0.76 | 0.16 | 0.95 | 0.81 |
| Herbs | 0.18 | **0.00** | 0.67 | 0.32 | **0.00** | 0.69 |
| Shrub leaves | 0.76 | 0.53 | 0.29 | 0.85 | 0.59 | 0.27 |
| Shrub stems | 0.86 | 0.67 | 0.34 | 0.91 | 0.70 | 0.32 |
| Soil Depth | $C_{burnt} < C_{control}$ | | | $N_{burnt} < N_{control}$ | | |
| | 12 | 28 | >44 | 12 | 28 | >44 |
| 0-5 cm | 0.75 | 0.84 | **0.02** | 0.43 | 0.84 | 0.03 |
| 5-30cm | 0.69 | **0.98** | 0.94 | 0.51 | 0.78 | 0.94 |
| 30-60 cm | 0.17 | 0.61 | 0.25 | 0.09 | 0.57 | 0.38 |





## 3.2 Fire influence on C and N cycling

Fire affected C and N concentrations and isotope ratios in aboveground biomass and soil (Fig. 2 A-H, Table S9, Table S10). While fire had generally strong effects on C and N concentrations as well as on isotope ratios in vegetation, only the upper soil layer showed differences in concentrations and isotope ratios. Overall, fire effects on C and N concentrations and isotope ratios were most pronounced in lichens.

### 3.2.1 N concentration and C/N ratios

Fire had a positive effect on N concentrations in lichens and a negative on N concentrations in graminoids and herbs (Fig. 2 C). Those effects disappeared in all groups after >44 years post fire. The N concentrations were reflected in the C/N ratios, which increased in graminoids and herbs but decreased in lichens (Fig. 2 E). While fire significantly increased C and N concentrations in the upper soil layer on the oldest fire scar, we could not detect a fire effect on the deeper soil layers (Fig 2 D, F).

### 3.2.2 $\delta^{13}$C

Lichens on all three fire scars were relatively depleted in $^{13}$C in comparison to unburnt plots (Fig. 2 G). This was also the case in graminoids on the two younger fire scars (Fig. 2 G). In soil, the abundance of $^{13}$C changed with time after fire. The upper soil layer of the youngest fire scar was enriched in $^{13}$C in comparison to unburnt plots (Fig. 2 H). This relationship turned with time since fire, and we observed a relative depletion of $^{13}$C in the soils of the oldest fire scar in comparison to unburnt plots.

### 3.2.3 $\delta^{15}$N

Fire increased the abundance of $^{15}$N in lichens (Fig. 2 I). This effect was still visible four decades after fire. We found no significant fire effects on $\delta^{15}$N in vascular plants.



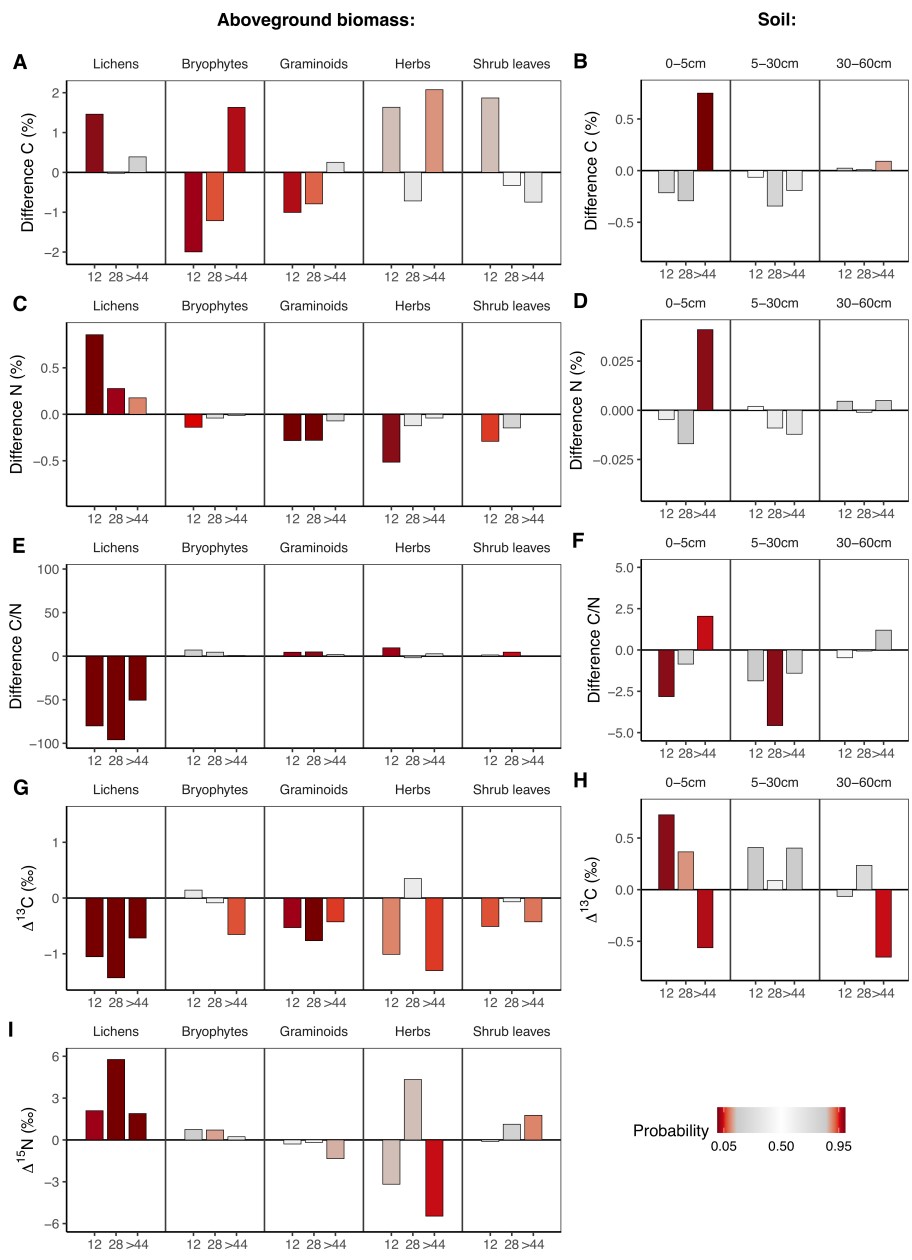

**Figure 2.** Difference (burnt - unburnt) between posterior means of C and N concentrations and isotope ratios in regard to time after fire in aboveground biomass and soil. Colour intensity indicates the probability P of the value from an unburnt control plot to be higher than the value from the burnt plot. If P is very high, there is a high probability of a negative fire impact. If P is very low, there is a high probability of a positive fire impact.



## 4 Discussion

Fire had no effects on total C and N stocks in subarctic dry tundra. This is probably because most of it was stored in soils (97.4% of C and 99.7% of N), which were largely unaffected by fire. In contrast, stocks in aboveground vegetation were
strongly reduced through fire, and the effect persisted even after more than 44 years. The lack of fire effects on carbon stocks seems to be contradictory to previous literature, as striking results reporting high losses of carbon through tundra fires (Mack et al., 2011) received a lot of attention and were covered by media (e.g. Rosen, 2017). However, negligible fire effects on carbon stocks have been reported previously for tundra ecosystems (Loranty et al., 2014). The effect of fire on tundra carbon and nitrogen stocks is variable and dependent on many factors. First, losses are strongly mediated by fire severity through
the consumption of organic matter (Schuur et al., 2003). Fire severity was relatively high for the fire investigated in Alaska (Mack et al., 2011) but intermediate in our study (Heim et al., 2021), which may explain the lack of a strong effect in our study (Maslov et al., 2020). Second, the thickness of the soil organic matter plays an important role. Because the decomposition of plant material is limited in tundra ecosystems by low temperatures and the often wet and anaerobic conditions, organic matter accumulates (Jonasson et al., 2001). A reduction of this soil organic layer leads to decreases in the amount of stored C (Mack
et al., 2004). Thus, exceptionally large losses of C and N stocks were reported from wildfires in rather wet tundra ecosystems with a thick soil organic layer or from peatlands (Mack et al., 2011; Turetsky et al., 2015). Our study was located in an upland area with well-drained mineral soils and a thin organic layer, and in such conditions, fires seem to cause only minor losses (Loranty et al., 2014). Further studies are needed to draw more general conclusions about the role of vegetation type and environmental settings (for example well drained upland soils) for wildfire-induced losses of C and N in tundra ecosystems. In
the long term, fire even increased C and N stocks in the upper soil layer, which was probably due to plant community turnover. The vegetation on burnt plots did not return to control levels within >44 years and was dominated by bryophytes, herbaceous plants, and shrubs, especially Betula nana (Heim et al., 2021). Vascular plants, and especially woody species, accumulate more soil C in roots and litter (Cahoon et al., 2016), and due to root exudates and their ability of symbiotic N fixation burnt plots can also contain more N than unburnt lichen dominated plots (Maslov et al., 2018). In contrast to soil, fire substantially reduced
C and N stocks in vegetation. Nevertheless, as vegetation stored much less C and N in comparison to soil, those effects were negligible for the estimation of total ecosystem stocks. In unburnt tundra, a large proportion of the aboveground C and N stocks was stored in reindeer lichens which dominated the vegetation before fire (Heim et al., 2021). As reindeer lichens recover very slowly, they were rare even several decades after fire (Frost et al., 2020; Heim et al., 2021) and did not reach the C and N stocks of unburnt sites. Instead, C and N were stored in shrubs and bryophytes, a pattern that reflects the vegetation recovery (Frost
et al., 2020). However, shrubs and bryophytes did not compensate for C and N loss in the burnt lichen biomass.

### 4.1 Fire affected long-term C and N cycling

Fire had a long-term effect on C and N concentrations and stable isotope ratios in aboveground vegetation, especially lichens, which suggests a strong, long-lasting effect of fire on C and N cycling in the ecosystem. The fire effects on nutrient cycling in soil were minor and most apparent in the topsoil.





### 4.1.1 No positive fire effects on the N concentration in vascular plants

Fire did not increase N concentrations in vascular plants. Originally, we expected that fire enhances mineralization rates, which releases previously unavailable N that can be taken up by vascular plants (Aerts, 2006; Salmon et al., 2016). However, our results do not support this hypothesis. On the contrary, we even found signs of nitrogen limitation in herbs and graminoids at the younger fire scars, as indicated by lower N concentration and higher C/N ratios in the biomass. This pattern may be linked to enhanced competition for nitrogen among vascular plants, which increased during post-fire succession (Bret-Harte et al., 2013; Heim et al., 2021). Unburnt plots were dominated by lichens, which obtain large parts of their nutrients from the atmosphere (Asplund and Wardle, 2017) and thus did not compete for available soil N. Therefore, vascular plants at unburnt plots may have relatively more available N. Lichens reflected strong long-term impacts of fire on N cycling. We found high N concentrations on burnt plots of the youngest fire scar. The disappearance of this effect with time since fire can be related to the fact that younger lichens generally contain more N (Kytöviita and Crittenden, 2007). Soil N concentrations were only increased on the oldest fire scar and in the upper soil layer. This pattern is less likely linked to the temperature-mediated increased microbial activity, as the soil temperature in the oldest fire scar recovered to control levels (Heim et al., 2021). Rather, this could be explained by the increased cover of vascular plants, which produce more root exudates, have symbiotic nitrogen fixation, and easily decomposable falloff litter (Maslov et al., 2018; McLaren et al., 2017).

### 4.1.2 Depletion of $^{13}$C in plant biomass and shifts in $\delta^{13}$C in soil

The relative abundance of $^{13}$C in soil changed with time after fire. We explain the higher soil $\delta^{13}$C in the burnt plots of the younger scar by higher decomposition rates and residual enrichment of $^{13}$C because of increased soil temperature after fire (Chambers et al., 2005; Jansson and Hofmockel, 2020). Decomposers prefer material less rich in $^{13}$C, which enriches heavy $^{13}$C in the remaining soil organic matter (Ehleringer et al., 2000). Contrastingly, the lower soil $\delta^{13}$C on burnt plots of the oldest fire scar (where soil temperatures returned to control levels) is probably linked to the altered vegetation composition with more shrubs and compound-specific variations of $\delta^{13}$C in litter. In plants, easily decomposing substances are comparatively enriched in $^{13}$C, while more refractory substances, such as lignin, are relatively depleted in $^{13}$C (Ågren et al., 1996). Increased shrub litter and a higher proportion of lignin at the oldest fire scar may thus result in a depletion of heavy $^{13}$C in the remaining organic matter of the soil (Ågren et al., 1996). The decreased $\delta^{13}$C in vascular and lower plants on fire scars in our study could also be related to lower $^{13}$C content of the $CO_2$ in the ambient air of fire scars (Dawson et al., 2002; Lakatos et al., 2007). Fire scars have higher temperatures and/or more easy degradable litter and thus a higher decomposition rate (Chambers et al., 2005; McLaren et al., 2017), which leads to increased soil respiration of $CO_2$ that is reduced in $^{13}$C in the course of decomposition (Farquhar et al., 1989).

### 4.1.3 No overall changes of $\delta^{15}$N during post-fire succession

We did not find an effect of fire on the abundance of $^{15}$N in the biomass of vascular plants. This was surprising because we expected that $^{15}$N in leaves of vascular plants on younger fire scars would be increased, while it would decrease in later





successional stages, through the progressing colonization with mycorrhizas (Hobbie et al., 2005). The lack of an effect of past fires on $\delta^{15}$N may indicate that the belowground mycorrhizas remained relatively unaffected by fire, and the resprouting strategy of tundra shrubs facilitates the resilience of dominant fungi against tundra fires (Hewitt et al., 2013). However, on

closer inspection of the shrub species Betula nana on the intermediate fire scar, we detected an enrichment of $^{15}$N in leaves of the burnt plots in comparison to plants on unburnt plots (Fig. S11). This indicates that the shrubs on the fire scar may not have been re-colonised again by mycorrhizas (Hobbie et al., 2005). We probably did not find an overall effect of fire on shrubs because we included too many species in the analysis that interact with different mycorrhizas or no mycorrhiza at all - which makes a pattern unrecognizable. While fire had no significant effect on $\delta^{15}$N in vascular plants, the positive effect of

fire on $\delta^{15}$N in lichens probably reflects the age of the thallus. Ellis et al. 2003 showed that there is a consistent pattern of $\delta^{15}$N distribution in the lichen thallus, with the highest $\delta^{15}$N content in the apices and a minimum located at 2-4 cm below the apex. The high $\delta^{15}$N can thus be related to the fact that lichen thalli on burnt areas are younger and thus smaller in comparison to unburnt stands (Heim et al., 2021).

## 5 Conclusions

While our results demonstrate that tundra fires do not generally reduce C and N stocks, we show that fire disturbance is an important long-term driver of C and N cycling in the subarctic tundra ecosystem of northern Siberia. We found that lichens play an important role in the storage of C and N in the aboveground biomass and that they strongly reflect environmental changes, such as increased soil respiration, after a fire. We did not find recovery to pre-fire conditions in terms of C and N cycling, even after more than 44 years. Our results suggest that ecosystem succession after fire may follow alternative pathways: While

N limitation and increased competition of vascular plants may lead the successional pathway back to a lichen dominated vegetation, the mycorrhizal re-establishment and thus an increased N availability for shrubs may constrain the first pathway and further strengthen the shrub dominance. Overall, our results strongly suggest that in lichen-dominated subarctic tundra ecosystems, the contribution of wildfires to the release of additional carbon to the atmosphere might be relatively small as soil stocks appear to be resilient.

## 270 6 Funding

RJH was supported by a scholarship from the Studienstiftung des deutschen Volkes. We acknowledge support from the Open Access Publication Fund of the University of Münster.

*Code and data availability.* The data and R-scripts that support the findings of this study are openly available in "Zenodo" at https://doi.org/10.5281/zenodo.5582683 and https://doi.org/10.5281/zenodo.5583511





*Author contributions.* RJH: Conceptualization, Data curation, Formal analysis, Funding acquisition, Investigation, Methodology, Project administration, Visualization, Writing – original draft preparation, AY: Data curation, Investigation, Methodology, Project administration, Resources, Validation, Writing – review and editing, AB: Supervision, Validation, Writing – review and editing, WH: Investigation, Validation, Writing – review and editing, VK: Investigation, KHK: Methodology, Resources, Validation, Writing – review and editing, CL: Formal analysis, Investigation, Validation, Writing – review and editing, AP: Investigation, DS: Investigation, FS: Investigation, NH: Conceptualiza-
tion, Funding acquisition, Methodology, Project administration, Resources, Supervision, Validation, Writing – review and editing

*Competing interests.* The authors declare that they have no conflict of interest.

*Acknowledgements.* We would like to thank Julia Aminova, Denis Bazyk, Leya Brodt, Marvin Diek, Bettina Haas, Liv Jessen, Olga Konkova, Ksenia Maryasicheva, Michael Moskowchenko and Daniel Rieker for help with data collection. We are grateful to the Department for Science and Innovation of the Yamalo-Nenets Autonomous Okrug (Alexei Titovsky), the Interregional Expedition Center "Arctic" (Elena Nyukina),
as well as to the Arctic Research Center of the Yamalo-Nenets Autonomous Okrug (Andrey Lobanov) for the great financial and logistic support and fruitful cooperation. For the trustful collaboration, we thank also Andrei Tolstikov (University of Tyumen). The C and N analyses of this study were carried out in the laboratory of the Institute of Landscape Ecology. The assistance of Ulrike Berning-Mader is greatly acknowledged.



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
