# Peer review of "Fire in lichen-rich subarctic tundra changes carbon and nitrogen cycling between ecosystem compartments but has minor effects on stocks"

_Biogeosciences, 2021_

## Author Response (AR1)

*We thank the Reviewer for the helpful and constructive comments helped to greatly improve our manuscript. We incorporated all raised points as suggested below and are confident that we can meet and successfully address all issues. Our answers and proposed text improvements are written in italic and bold below each comment.*

**Anonymous Referee #1**

This study presents an analysis of post-fire carbon (C) and nitrogen (N) dynamics using a chronosequence of tundra ecosystems in western Siberia. Results show minimal effects of fire on above and belowground nutrient pools, with the primary effect related to lichen declines and bryophyte increases post-fire. The research is well designed with appropriate analyses, and the manuscript makes an important contribution to the increasingly important topic of tundra fires for a region that is not well documented in the English language literature. Several aspects related to the framing and interpretation of the work may help to improve manuscript. Congratulations overall on a nice study.

*Thank you!*

Framing this work as contradictory to Mack et al (2011) doesn't seem entirely correct to me and raises a couple of important issues to consider. The paper by Mack estimates combustion losses one year post fire by using proxies to reconstruct the organic soil layer. In the absence of such data, the present study has a slightly less complete picture of soil C losses.

*This is true. We removed the text part that presents our results as contradictory to Mack et al.. We now relate our results to Mack et al. by referring to the different time frames of the two studies. While Mack et al. 2011 is mainly about the combustion loss, we investigated the long-term fire consequences on C stocks. Therefore, we stated right at the start of our manuscript that we do not look at combustion losses, but the long-term consequences. In the first sentence we changed "effects" to "long-term effects" and maintained this phrasing throughout the manuscript.*

The data provide good information on changes in C concentration, but I wonder if the depth increments are comparable. For example, could it be the case that 0-5cm in the burnt sites corresponds to 10-15cm in the unburned sites?

*In contrast to Mack et al. we worked on rather fine-grained mineral soil without significant organic layers that could combust, which is typical for the dry and lichen-rich tundra type. A thick organic layer with accumulated litter, as described in Mack et al. 2011, was not present in our sites. Instead, the mineral soil was covered by living lichen and moss in unburned sites, lying directly on the mineral layer.*
*The 0-5 cm layer in our study is the upper layer of the mineral soil, which was in unburned sites found directly under the lichen layer. The mineral horizon was always clearly identifiable and could not be misinterpreted.*
*In a freshly burned site, which we also visited, the organic layer (lichen, moss and roots) was not entirely burned. But this likely corresponds to the rather small fire scar size in this case.*

These things are OK, but some more details and nuanced discussion would help. For example, what are the soils like at these sites, is there a well developed organic layer?

*We added to the discussion in l. 228ff:*

*"Thick organic soil layers, as e.g. reported for a site in Alaska (21.5 cm, Mack et al., 2011) may store particularly large amounts of C and N that are potentially lost if burnt. However, much shallower organic layers have been reported for North-Eastern Siberia (6.3 cm, Loranty et al., 2014) and were also typical of our study area (average thickness 4.3 cm). Therefore, potential losses upon burning can be expected to be much lower, and, moreover, the importance of the more stable mineral soil carbon stock less susceptible to losses from burning relatively increases. However, a direct comparison of these studies is not possible, as we investigated the system during regeneration and not directly after combustion."*

Would such combustion losses be possible, or are soils a less important fuel source in these systems?

*In the subarctic dry tundra, as studied here, we would assume potential combustion losses to be generally smaller, as the soil organic layer is thin, and carbon stored in the mineral soil is much more stable. We discussed this point as shown in the answer to the next comment.*

*Additionally, regeneration was possible in all sites, in our study. What we show are the difference between burned and unburned sites after different times of recovery, this means new vegetation and hence a regeneration of the carbon stocks.*

Note that Loranty et al (2014) report organic soil depth in this context, and that Jones et al (2013) also document organic soil accumulation after fire. If these sites have different soils it may be worth highlighting this and considering what this means for geographic variability in tundra fire impacts on biogeochemical cycling.

*Good idea! We discussed the impact of geographical variability in soils on the biogeochemical cycling as follows after line 215:*

*"Our results support recent findings of (He et al., 2021), who claim that soil organic carbon loss through tundra fires may be overestimated because the soil organic layer in many tundra areas is much lower (see also our findings) than in the area of the extreme Anaktuvuk River Fire with predominant wet organic soils (Histosols).*

*In addition to the thick organic layer which usually is combusted by fire (He et al., 2021) moist tundra ecosystems typically store much higher amounts of carbon per unit area because of a higher soil organic matter content in comparison to dry tundra ecosystems (Marion et al., 1997). Through increased temperature sensitive decomposition after fire, (De Baets et al., 2016) burning can thus indirectly lead to higher decomposition rates and ecosystem carbon loss in wet tundra ecosystems. Depending on tundra type, fire impacts can thus have varying severity of fire impact on biogeochemical cycling.*

Related to these points, the results or discussion don't really mention depth differences for the soils, particularly regarding the 13C results. Could there be differences related to belowground biomass (root) dynamics? Alternatively, could differences in permafrost thaw depths or temperatures explain any of these differences?

*We discussed our observation that $^{13}C$ differed between the fire scars and the unburned control plots in the topsoil layer in section 4.1 of the discussion. We reasoned that these differences may be related to differences in the decomposition rate caused by higher soil temperatures at the fire sites, not as a direct result of the fire, but an indirect due to the thinner vegetation layer shielding the soil surface less from sunlight. Since temperature decreased rapidly with depth, primarily the topsoil layer was affected. However, we do not think that this finding helps explaining the differences between the dry Tundra ecosystem in our study and the wet Tundra in Alaska.*

Here again I think some site specific context would be helpful - could results from your other studies at these sites (e.g. soil temperature or thaw depth) help interpret these results?

*We added means of active layer depth and soil temperature for each fire scar and control plots to the study site description in lines 83ff.*

A map and/or photos of the study sites could be helpful as well.

*We added a detailed image of the study site with the fire scars to the study site description.*

This may be more personal preference, but I think the structure of the manuscript could be improved. Subheadings seem to be used instead of paragraph breaks. The Introduction should be broken into several paragraphs to help highlight main aspects of the topic. Conversely, there are places where the subheadings seem excessive - for example section 3.2 could be a paragraph or two.

*We agree that this was not ideal. We introduced more paragraphs in the Introduction section, as proposed by your comments below, and we reduced the number of subheadings in the Results section.*

Presenting these results in a bit more detail and narratively linking them can help provide a more comprehensive overview for the reader.

*Thank you we agree and revised the results in the revision focussing on improving the narrative.*

L30: Perhaps start a new paragraph when switching from C to N.

*Inserted!*

L40: New paragraph?

*Inserted!*

L186: This discussion begins to address some of my points above. Note the study by Loranty et al had ~10cm organic soil layer relative to ~21cm reported by Mack et al., and in both cases the boundary between organic and mineral horizons is generally well defined. It would be interesting to know how the sites in this study compare, and whether differences in soil types between depth layers (i.e. 0-5cm, 5-30cm affect bulk density and nutrient concentrations.

*We added more detailed information regarding soil organic layer depth to the discussion (please see answer to comment above).*

**Anonymous Referee #2**

The manuscript by Heim et al. reports on C and N stocks and their stable isotope composition in various ecosystem compartments (soil organic matter, aboveground vegetation including lichen and vascular plants) in three large fire scars (between 12 and 44 years post-fire) and adjacent control sites; in Western Siberia. Only d15N data on soil samples are missing (due to low soil N concentrations, although I would think this is quite feasible from a technical point of view).

The authors' main conclusions are that

(i) total ecosystem C and N stocks were not significantly affected by fire.

(ii) soil C and N stocks were not affected. Soil make up the majority of ecosystem C and N stocks.

(iii) vegetation was affected, mainly due to a reduction in the lichen layer, which takes a long time to recover.

(iv) a few trends in d13C and d15N – but those are not well explained, see further.

Personally, I would stick to the first 3 (which are related, obviously). Given that the soils in these systems are relatively poor in organic C (from less than 1 to slighty over 2% on DW basis), these conclusions are in line with expectations. The interpretation of the isotope data is not conclusive and too speculative.

*Thank you for this helpful assessment - more on this point in the answers to the following comments!*

Overall, I would mostly encourage the authors to take out the more speculative sections, keep the manuscript and conclusions to what can be unambiguously demonstrated, i.e. fire affects aboveground biomass, and not belowground OC stocks in a system with rather mineral soils, and as the biomass presents a small fraction of ecosystem OC stocks, those are not strongly affected. The isotope data are intriguing, but I do not feel much can be drawn from them at this stage.

*Good point. The fact that there is hardly any isotope data regarding fire effects in tundra ecosystems available makes a discussion quite difficult, we acknowledge this and therefore keep this section now much shorter. However, we are convinced that our data offers potential for discussions and interpretation in future studies of this topic and would thus prefer to leave it in. Nevertheless, we agree to largely follow your suggestions in substantially shortening these sections and limiting our interpretation.*

Carbon stable isotope data

-abstract L 11-13: "This could be related to …": this is not conclusive, and your data do not really allow you to draw anything firm from this.

*We removed this conclusion.*

-Figure 2: use small delta symbol, not capital delta on y axis label

*We present differences in the "small" delta on the y axis ($\Delta = \delta_2 - \delta_1$) to facilitate identifying differences. In this case to our knowledge a capital delta is commonly used.*

-The mechanism behind the decrease in d13C in plants and lichen is not well understood. I am not very convinced on the suggestion that lower d13C in local atmospheric CO2 due to the fire would be the cause. This would imply a higher CO2 concentration, from increased mineralization – but the soil OC stocks suggest there is no enhanced mineralization (at least not observed as a decrease in C stocks). Such local variations in CO2 and d13C-CO2 are typically observed in closed canopy systems. Without actual data demonstrating local gradients in d13C-CO2, I would not make this suggestion too explicit. The authors refer to Dawson et al. (2002) and Lakatos et al. (2007) in this context – but neither of these papers mention anything about fire and its effect on local d13C-CO2 years after the actual burning.

*Yes, this is a good point as well. We discussed those ideas more carefully, now. The text in lines 273ff is now:*

*"Why $\delta$13C in vascular and lower plants on fire scars is decreased in our study remains relatively unclear as variations in $\delta$13C are usually complex and not straightforward to interpret (Dawson et al., 2002). Therefore, our findings could not be related with certainty to a process described in literature. One reason for the decreased $\delta$13C might be the lower 13C content of the CO2 in*
*275 the ambient air of fire scars. A lower 13C content of the CO2 can be explained by increased decomposition rates (Dawson et al., 2002; Lakatos et al., 2007). However, we could not detect a decrease in C stocks in our data that would allow the assumption of increased mineralisation."*

Alternative explanations for minor shifts in plant d13C (water availability or sources, interactions with nutrients, ..) are not considered (although with the data at hand, one would not be able to make a strong case for a precise mechanism).

*Yes, we thought about other alternative explanations as well, however, it seems that the knowledge on these complex relationships seems to be relatively small for further reasonable explanations. We stated in l. 178ff*

*"In our case, water availability may be a better explanation for the observed shifts towards lower 13C values in vegetation, as increased water availability is common in post-fire permafrost landscapes (Holloway et al., 2020). This is in line with the*
*280 frequently observed negative correlations of indicators for humidity and tree ring 13C (e.g. Holzkämper et al., 2012), as 13C reflects stomatal conductance as affected by moisture availability or drought stress. Dawson 2002 states that "But unlike in vascular plants, delta tends to increase with water limitation in nonvascular plant taxa (Williams and Flanagan 1996, 1998)". Our soil moisture data, however, does not support this, which might be due to the timing of sampling (soil moisture is also probably the most variable parameter in space and time)."*

-Hence, I also find that conclusions such as "they [=lichen] strongly reflect environmental changes, such as increased soil respiration, after a fire", since the data presented do not show direct evidence for increased soil respiration.

*Yes, this is a good point. We removed this rather uncertain discussion from our conclusions.*

-The explanation offered for the soil d13C data is also too speculative in my opinion. Linking such a small difference to increased temperatures (no data are presented to back up an increase in temperature), and I feel that Ehleringer et al. (2000) is not adequately interpreted here: microbial communities do not have an inherent preference for 13C-depleted organic matter- that's not what Ehleringer et al. conclude. Microbial biomass appears to be slightly enriched in 13C, yes – but again, given that the soil C stocks do not appear to change, why invoke higher mineralization (and thus, a higher contribution of microbial biomass) ?

*Yes, thank you again for expressing your concerns. The explanation is indeed quite speculative, and we removed it from the discussion.*

Nitrogen stocks
The scenarios described in Section 4.1.1. are quite speculative – possibilities, but nothing conclusive.

*Yes, we discussed this part more cautiously in l. 256ff:*
*"A possible explanation for this pattern may be linked to enhanced competition for nitrogen among vascular plants, which increased during post-fire succession (Bret-Harte et al., 2013; Heim et al., 2021). Unburnt plots were dominated by lichens, which obtain large parts of their nutrients from the atmosphere (Asplund and Wardle, 2017) and thus did not compete for available soil N. Therefore, vascular plants at unburnt plots may have relatively more available soil-N. Lichens reflected long-term impacts of fire on N cycling. We found high N concentrations on burnt plots of the youngest fire scar. The disappearance of this effect with time since fire might be related to the fact that younger lichens generally contain more N (Kytöviita and Crittenden, 2007).*
*Soil N concentrations were only increased in the upper soil layer of the oldest fire scar. This pattern is less likely linked to the temperature-mediated increased microbial activity, as the soil temperature in the oldest fire scar recovered to control levels (Heim et al., 2021). Rather, this might be explained by the increased cover of vascular plants, which produce more root exudates, have symbiotic nitrogen fixation, and easily decomposable falloff litter (Maslov et al., 2018; McLaren et al., 2017)."*

Statistical analysis: The approach used by the authors is outside my comfort zone, but I do not understand why they follow this road (posterior distributions). It would be good to shed some light on why this is helpful, and why not simply use the actual measured data. I would personally prefer to see the actual data presented in the ms, and the predicted values in the supplement.

*We used data analysis in a Bayesian framework, as the sample size of our study is too small to include several random factors which we have to include to correct for the study design. On the base of the measured data, we can therefore not state anything regarding significance.*
*The raw data is shown in the appendix and can be also downloaded via: https://doi.org/10.5281/zenodo.5582683*

Minor suggestions
-species names should be in italics throughout
*Thanks. Corrected throughout.*

-soil d15N data: given the N concentrations (0.05 % DW approximately), I don't see why this is not possible – 20 mg of dried sample should provide 10 µg N for analysis, which is largely sufficient for good d15N data.
*We did several tests to capture a reliable $^{15}N$ signal. However, the very large masses of sample needed to obtain good $^{15}N$ data (indeed 20-30 mg) caused several problems. Firstly, large tin capsules got stuck in the autosampler, which made manual sample insertion necessary. After solving this it turned out that such large sample masses lead to very high amounts of ash residues in the combustion tubes. This affected the quality of the combustion significantly, causing higher yields of NOx which could not fully be reduced back in the reduction reactor. Therefore, $^{15}N$ values of these samples had very bad reproducibility and made very frequent cleaning of the oxidation reactor necessary. Due to the insufficient quality of data and as little data is available from other studies to provide room for comparison and interpretation, we decided to skip $d^{15}N$.*